# Electrochemical Synthesis of Nanocrystalline CuAg Coatings on Stainless Steel from Cyanide-Free Electrolyte

**Manal A. El Sayed [1], Magdy A. M. Ibrahim [2,*], Nahla T. Elazab [3] and Malek Gassoumi [4]**

[1]   Department of Physics, College of Science and Arts, Qassim University, Bukairiayh 51452, Saudi Arabia
[2]   Department of Chemistry, Faculty of Science, Ain Shams University, Abbassia, Cairo 11566, Egypt
[3]   Department of Biology, College of Science, Qassim University, Buraydah 51452, Saudi Arabia
[4]   Department of Physics, College of Science, Qassim University, Buraydah 51452, Saudi Arabia
*   Correspondence: magdyibrahim@sci.asu.edu.eg

**Abstract:** Herein we demonstrate a novel plating bath, free from cyanide, to plate a highly adherent nanocrystalline copper-silver (ncCuAg) coating on a stainless-steel substrate and its application as an antimicrobial coating. The microstructures, such as the grain size, texture, microstrain, and the crystalline preferential orientation of CuAg deposits, are systematically investigated by X-ray diffraction analysis. The range of 13.4–16.6 nm was discovered to be the crystallite size determined from the X-ray peak broadening (Scherrer's formula). Both HRTEM, FESEM-EDS, XPS, and mapping analysis revealed that the ncCuAg coatings are composed of both Ag and Cu atoms. Electrochemical processes occurring during CuAg co-deposition were investigated by using linear sweep voltammetry (LSV), cyclic voltammetry (CV), and anodic linear stripping voltammetry (ALSV). Additionally, the coatings made of ncCuAg produced by these baths work well as antibacterial agents against gram-positive (*Staphylococcus*) and gram-negative bacteria (*Escherichia coli*).

**Keywords:** ncCuAg coatings; electrodeposition; antimicrobial effect; cyanide-free electrolyte; stainless steel; HRTEM

## 1. Introduction

During the COVID-19 outbreak, antimicrobial technologies are in high demand for use in hospitals, schools, and other high-traffic areas. Antibacterial properties are found in several metals, but their potential advantages must be weighed against their toxicity. By using element alloying, appropriate metal forming, and heat treatment, antibacterial metallic materials can effectively suppress bacterial adhesion, growth, and proliferation [1]. Antibacterial metal alloys containing Cu and Ag have been reported to have high antibacterial activity against several types of bacteria, such as antibacterial stainless steel [2]. A newly designed CuAg coating has been examined and proven to be highly effective against MSSA and MRSA, Pseudomonas aeruginosa, and Enterobacter aerogenes [3]. Copper, silver, and their alloys are also non-toxic and can be used effectively to control bacterial growth. Due to their inherent antibacterial qualities, metallic copper and copper alloys are perfect for situations where other metals fall short [4,5].

The development of antibacterial stainless steel, which provides a sanitary and clean surface to avoid microbial diseases, is one of the most appealing aspects. This type of material is in high demand due to rising public concern and the spread of antibiotic-resistant microorganisms. Antibacterial stainless-steel products have a wide range of applications, including hospitals and the food processing industry [6]. Antibacterial stainless steel is traditionally made by alloying it with copper or silver. The issue with copper-alloyed stainless steels is that they are difficult to maintain both antibacterial and corrosion resistance at the same time [7].

Silver containing stainless steels demonstrated antibacterial effects against bacteria such as MRSA, Salmonella, *Escherichia coli*, and *Staphylococcus aureus* in an antibacterial test

conducted by Yokota et al. [8]. Such silver-alloyed stainless steels have not been proven commercially successful in the market, owing to their high cost. A brand-new strategy involves depositing a thin film over stainless steel to create antibacterial characteristics on the surface. Although much has been written about the preparation of noble metal alloys, there are few reports on bimetallic copper alloys [9,10], particularly with silver [11]. Cu and Ag have very different lattice constants (0.409 and 0.361 nm for Ag and Cu, respectively), making the manufacture of their alloys problematic. Moreover, because of the difference in redox potential, it is challenging to manage the simultaneous reduction of Cu and Ag [12]. A further big obstacle is copper's instability in an aqueous medium.

The synthesis of CuAg was achieved using different techniques such as laser cladding [13], blasting [1], microwaves [14], ball milling [15], chemical vapour deposition [16], pulsed laser deposition [17], wet chemical method [18], ion beam mixing [19], and electrodeposition (ED) [7,20]. The ED technique has many potential advantages over the other techniques mentioned above, including the ability to deposit nanocrystalline coatings at a low cost, the ability to produce compact pore-free dense coatings, crystal particle size, microstructure, and roughness [21–23]. These can all be easily controlled. In addition, plating on any complex shape for antimicrobial touch surface application is possible. Moreover, in ED, the additional energy required to generate the metastable solid solution is typically less than 1 eV per atom [24]. The ED of CuAg was carried out primarily from cyanide baths [25,26]. Many researchers are looking for cyanide-free baths due to the limitations of employing cyanide ions in the industry. Some work focuses on acidic sulphate baths [27,28], ammonia solution [29,30], hydrazine sulphate [31], methanesulfonic acid baths [32], protic ionic liquid [33], and pyrophosphate-iodide electrolyte [24]. However, the results are not encouraging. For example, the adhesion of CuAg obtained from ammonia solution has not been established, only powders [29]. Furthermore, the conditions of dendritic or nodular cluster growth remain unknown. However, one study reported that a CuAg with 10% Ag created by laser cladding copper and silver on stainless steel had stronger biocidal activity than the pure components against *Escherichia coli* [13]. As a result, the current research intends to introduce a novel bath free from cyanide ions for the ED of highly adhered and compact ncCuAg coatings on stainless steel surfaces to prevent germs from spreading. Scanning electron microscopy (SEM), energy dispersive X-ray spectroscopy (EDS), transmission electron microscopy (TEM), and X-ray diffraction (XRD) analysis were used to assess the coating microstructure. Measurements with a linear sweep and cyclic voltammetry were used to determine some characteristics of the electrochemical behavior of the electrolyte.

## 2. Experimental

### 2.1. Electrochemical Synthesis of ncCuAg Coatings

The ED of ncCuAg coatings from cyanide-free solutions was investigated using the novel electrolytic bath shown in Table 1. The copper sulphate ($CuSO_4 \cdot 5H_2O$), and the silver nitrate ($AgNO_3$) were used as metal sources. Citric acid ($C_6H_8O_7$) was used as a complexing agent; sodium sulphate ($Na_2SO_4$) was used as a supporting electrolyte; sodium hydroxide (NaOH) was used to form sodium citrate; nitric acid ($HNO_3$) was used to prevent the precipitation of M-citrate (M: $Ag^+$ or $Cu^{2+}$); and polyethylene glycol (PEG) was routinely used as a suppressing agent [34–36]. This bath is characterized by its cyanide-free, high stability, working at ambient temperature (27 °C), and the electrolyte was very clear for several months without any precipitation. Moreover, the adhesion of the ncCuAg co-deposited on the SS surface from this bath was very good without using Ni-strike before the deposition process [37].

**Table 1.** The optimum bath composition for the ncCuAg co-deposition and the operating conditions.

| Substance | Concentration/(M) |
|---|---|
| $AgNO_3$ | 0.0125–0.1 |
| $CuSO_4$ | 0.5 |
| $C_6H_8O_7$ | 0.1 |
| NaOH | 0.1 |
| $HNO_3$ | 0.05 |
| $Na_2SO_4$ | 0.14 |
| PEG | 2.0 g/L |
| Electrodeposition conditions | |
| Current density (mA cm$^{-2}$) | 1.14, 1.72, 2.29, 3.43 mA cm$^{-2}$ |
| Deposition time (min) | 10 min |
| Temperature (°C) | 27 |
| pH | 1.9–2.4 |
| Cathode: | Stainless steel 304 |
| Anode: | Pt sheet |

*N.B.* **Bath I** refers to a bath containing: 0.0125 M $AgNO_3$, 0.5 M $CuSO_4$ and the rest of the other constituents in the Table.

The ED of pure Cu from this bath (in the absence of $AgNO_3$) was carried out for comparison. In addition, our workgroup successfully prepared nanocrystalline silver coatings on SS using the same bath constituents (without $CuSO_4$) [38]. The bath composition of 0.0125 M $AgNO_3$ and 0.5 M $CuSO_4$ with the rest of the constituents in Table 1 was used as an optimum bath to produce the ncCuAg coatings and is denoted as bath I.

During the ED, a flat sheet of stainless steel (3.5 × 2.5 cm) was employed as a cathode, and a platinum sheet of the same size was utilized as an anode. The ED was carried out in a non-stirred environment at an ambient temperature. The stainless steel used and the process of its cleaning were described elsewhere [38]. The ncCuAg coatings were electroplated galvanostatically (at constant current density) for 10 min. A glassy carbon electrode (GCE) with an area of 0.1963 cm$^2$ was used for the voltammetric tests. A double junction silver–silver chloride electrode was employed as a reference electrode. As an auxiliary electrode, a Pt wire was employed. The electrochemical measurements were performed using an interface 1000 Instrument (Warminster, PA, USA) Potentiostat ZRA.

Energy dispersive spectroscopy (EDS) was used to assess the alloy composition using a detector coupled to a JEOL JSM-6700F field emission scanning electron microscope (FE-SEM). Surface morphology was characterized by using SEM. The size of the synthesized alloys was determined using the ImageJ software. Using a Panalytical X'Pertdiffractomer with Cu K radiation (λ = 1.5418) in the θ/2θ geometry, the crystal structure was determined using X-ray diffraction. The produced samples were evaluated using the Thermo Scientific K-Alpha XPS equipment (Waltham, MA, USA). To extract the chemical state information, the sample was irradiated with a monochromatic Al K X-ray source, and for high-resolution spectra, the analyzer passed an energy of 200 eV with a step size of 1 eV. The adhesion of ncCuAg coatings was carried out using a cross-hatch adhesion tester/Elcometer 1542. Cutters, such as the Elcometer 1542, have a wheel at the opposite end to the cutter. When placed on the surface, the applied load is distributed evenly both along and across the handle, ensuring consistency in the method. A cut piece of tape should be placed over the test area and smoothed down firmly using a fingernail or fingertip to ensure good adhesion between the tape and the coating. After that, the tape is removed from the coating. The standard being applied determines the kind of tape and the angle at which the tape is removed. Then, the evaluation of the coating determines the degree of coating removal in accordance with the protocol (ISO or ASTM). The ISO 0–5 represents best

to worst, whilst the ASTM 5B-0B represents best to worst. Adhesion characterization quantitively of the deposited ncCuAg was performed using the Pull Off Adhesion Tester. The Positest Adhesion Tester (Model AT-M) is designed to measure the bond strength of the applied coatings.

### 2.2. Antibacterial Activity Measurement

### 2.2.1. Organisms under Investigation

*Escherichia coli* as a gram-negative enteric pathogen and *Staphylococcus aureus* as gram-positive bacteria were isolated and identified from pathogenic samples using a standard laboratory protocol, cultured on suitable media, and purified as described by [38]. The two bacterial strains were grown for 24 h at 37 degrees Celsius in nutrient broth on a rotary shaker (180 rpm). They were then subcultured until their optical densities reached 0.02–0.05, as determined by a spectrophotometer at 610 nm, in order to be used in the next assay.

### 2.2.2. Assay for Agar Diffusion

The ncCuAg samples were examined for their antibacterial activity by the agar diffusion method. A total of 100 mL of the prepared bacterial suspension was spread randomly on a sterilized solid nutritional medium under control conditions. Then, the inoculated agar was used to hold the surface sterilized alloy samples. To allow alloy particles from the samples to permeate the agar, the inoculation plates were refrigerated for two hours at 4 °C before being incubated for 24–48 h at 37 °C. The clear zone around the samples was measured in cm.

## 3. Results and Discussion

### 3.1. The Electrochemical Synthesis of the ncCuAg Coating

The ED of ncCuAg from the cyanide-free electrolyte is practically impossible due to the far potentials of the parent element as well as the complicated chemistry of the bath. Furthermore, because the CuAg system is totally thermodynamically immiscible at ambient temperature, the overall atomic contact between Cu and Ag is repulsive, preventing underpotential deposition [39]. Preliminary experiments were carried out to choose the suitable operating conditions such as temperature, current density, and plating time. The impact of temperature was investigated between 27–60 °C. Room temperature at 27 °C was ideal because it results in a smooth and satisfying deposit. Some surface pits are visible, and the deposit begins to darken as the temperature rises. On the other hand, it was found that 10 min and 1.72 mA cm$^{-2}$ were the suitable operating conditions for producing sound and satisfactory alloy deposits. Therefore, the electrodeposition of the CuAg was carried out galvanostatically on the SS surface for 10 min. at room temperature (27 °C) and at 1.72 mAcm$^{-2}$ as an optimal condition. Moreover, it was found that the ionic ratio [Ag$^+$]/[Cu$^{2+}$] in the bath is very critical. As a result of preliminary experiments, it was discovered that an ionic ratio [Ag$^+$]/[Cu$^{2+}$] in the range of 0.025–0.2 produces a high-quality ncCuAg with Ag content ranging from 2.17 to 6.59%(wt.%) depending on the operating conditions. The operating conditions of 1.72 mA cm$^{-2}$, 10 min, 27 °C produced the ncCuAg that contained 6.59% Ag. This conclusion contradicts the notion that the CuAg system's equilibrium binary phase diagram shows that silver solubility in a bulk copper matrix at ambient temperature is less than 0.08 at. percent [40].

Although the potential of Ag/Ag$^+$ is nobler than that of Cu/Cu$^{2+}$, the copper content within all the samples is significantly higher than Ag. Because Ag$^+$ ions are present at a significantly lower concentration (0.0125 M) than Cu$^{2+}$ ions (0.5 M), this finding is considered to be connected to the mass-transfer limitations within the system. As a result, Ag$^+$ ions are consumed more quickly by the redox replacement process near the electrode, whereas the effects of the mass-transfer limit on Cu ions are less severe, allowing for a higher degree of Ag replacement by Cu on the electrode. When citric acid or other hydroxycarboxylic acids are used to dissolve AgNO$_3$ and CuSO$_4$, the precipitation of silver citrate (Ag$_3$C$_6$H$_5$O$_7$) and copper citrate (Cu$_3$(C$_6$H$_5$O$_7$)$_2$) does not occur [38]. The following

reaction can be used to explain this phenomenon for silver nitrate, copper sulphate, and citric acid:

$$3AgNO_3 + H_3C_6H_5O_7 = Ag_3C_6H_5O_7 \downarrow + 3HNO_3 \tag{1}$$

$$3CuSO_4 + 2H_3C_6H_5O_7 = Cu_3(C_6H_5O_7)_2 \downarrow + 3H_2SO_4 \tag{2}$$

Consequently, the equilibrium of reactions (1) and (2) might be assumed to have shifted to the left.

### 3.2. Electrochemical Studies

Typical LSV for Ag, Cu, and CuAg reduction at GCE from their corresponding electrolytes (Table 1) are given in Figure 1. For the ED of Cu alone, $AgNO_3$ is absent from the bath in Table 1, and for the ED of Ag alone, $CuSO_4$ is absent from the bath in Table 1. Linear sweep voltammetry was used to establish the potential range for Cu and Ag deposition. The LSV for Ag alone exhibits a small peak at 0.38 V in accordance with the reduction of $Ag^+$ ions followed by a current plateau. The current corresponding to the $Ag^+$ ion reduction is small as a result of its low concentration (0.0125 M). However, the deposition of Cu (with certain hydrogen evolution) started at 0.02 V, followed by a sharp increase in current as a result of the high $Cu^{2+}$ ion concentration in the bath. The addition of $Ag^+$ ions to the electrolytic solution containing $Cu^{2+}$ ions leads to acceleration of the Cu deposition, as is obvious from the positive potential shift in the LSV of the alloy curve and the higher current density obtained. This means the addition of $Ag^+$ ions have a depolarizing effect on the copper ED, in accordance with the work of Shao et al. [28].

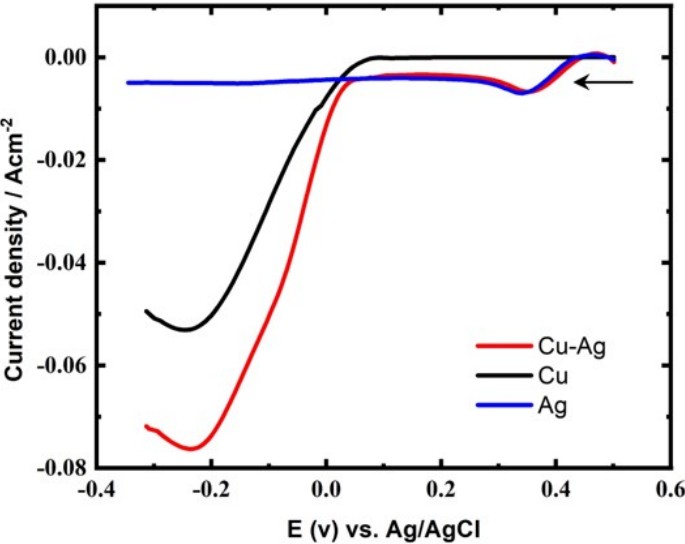

**Figure 1.** LSVs for Ag, Cu and CuAg reduction at GCE, sweep rate of 5 mV s$^{-1}$, and at 27 °C.

Figure 2 illustrates the LSV for the CuAg deposition with constant $CuSO_4$ concentrations (0.5 M) and various concentrations of $AgNO_3$ (0.0125–0.1 M). LSV is characterized by the presence of a reduction peak at 0.38 V (attributed to the silver reduction) and a current plateau followed by a sharp increase in current. The reduction peak as well as the current plateau increase with increasing the $Ag^+$ ion concentrations. This finding shows that raising the content of $Ag^+$ ions in the electrolytic solution enhances the rate of alloy deposition. This is because the deposition of Ag in the potential region of interest is controlled by diffusion, whereas the deposition of Cu is controlled by charge transfer.

Typical CVs for Ag, Cu, and CuAg recorded on GCE are shown in Figure 3. The CV for Ag consists of a fast reduction peak at 0.32 V. Ag dissolution occurs at a potential near to that of its reduction in the anodic scan, indicating that Ag ED is highly reversible. A nucleation loop is observed between the oxidation and reduction curves (see the inset of Figure 3), which is typical of ED systems [41]. For the Cu solution, the $Cu^{2+}$ ion is reduced

on GCE, starting at 0 V, and followed by a diffusion-limited reduction peak at −0.45 V. On the reverse scan, the anodic dissolution of copper gives a peak at ~0.7 V. For the CuAg, the CV exhibits two cathodic peaks, I and II, corresponding to the $Ag^+$ and $Cu^{2+}$ ions reduction, respectively. The CuAg solution closely resembles the properties of elemental electrolytes, with Ag and Cu deposition and dissolution maxima occurring at the same potentials. The fact that the onset of Ag and Cu deposition occurs at the same potentials as in single metal solutions involves that the two ionic and atomic species have no interaction [42]. This also shows that Ag and Cu are deposited separately, resulting in the formation of separate phases, as stated by Liang et al. [43].

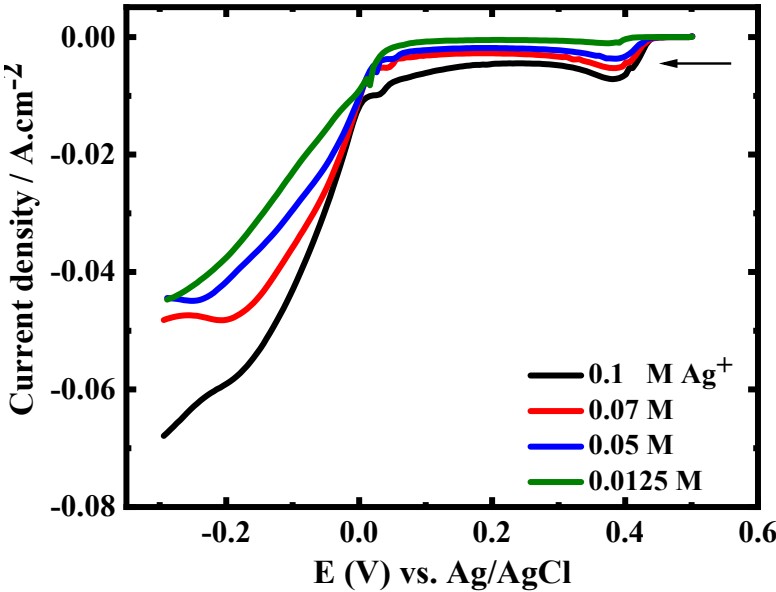

**Figure 2.** LSVs for CuAg reduction with constant $Cu^{2+}$ ions concentration (0.5 M) and varying concentrations of $Ag^+$ ions, at GCE, 5 mV s$^{-1}$, and at 27 °C.

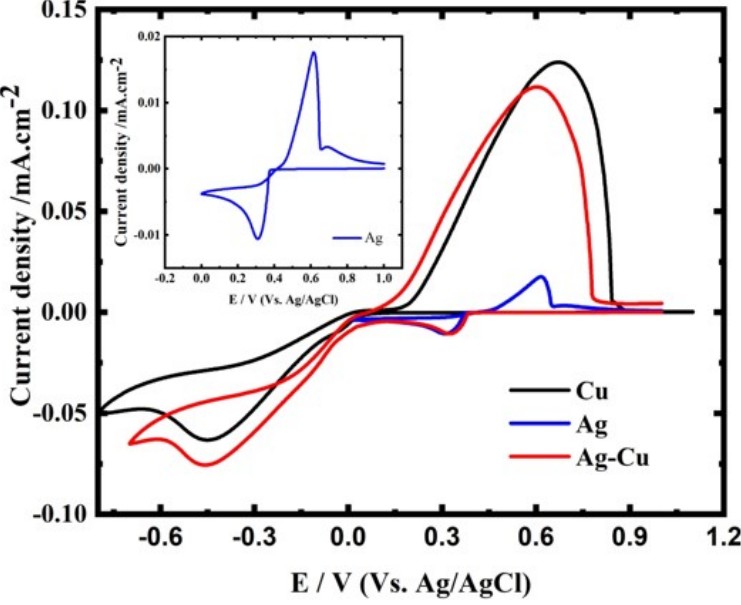

**Figure 3.** Typical CV of Ag, Cu, and CuAg recorded at the GCE, 50 mV s$^{-1}$, and at 27 °C.

The typical CVs for CuAg at constant $Cu^{2+}$ ion concentration (0.5 M) and varying contents of $Ag^+$ ions are explored in Figure 4. The CV shows two cathodic peaks, I and II, which are related to the reduction of $Ag^+$ ions and $Cu^{2+}$ ions, respectively. In the anodic scan, one oxidation peak is obtained, corresponding to the dissolution of the CuAg. The reduction peak I is enhanced with enhanced $Ag^+$ ion concentrations, while the height of the dissolution peak decreases as the $Ag^+$ ion concentration increases.

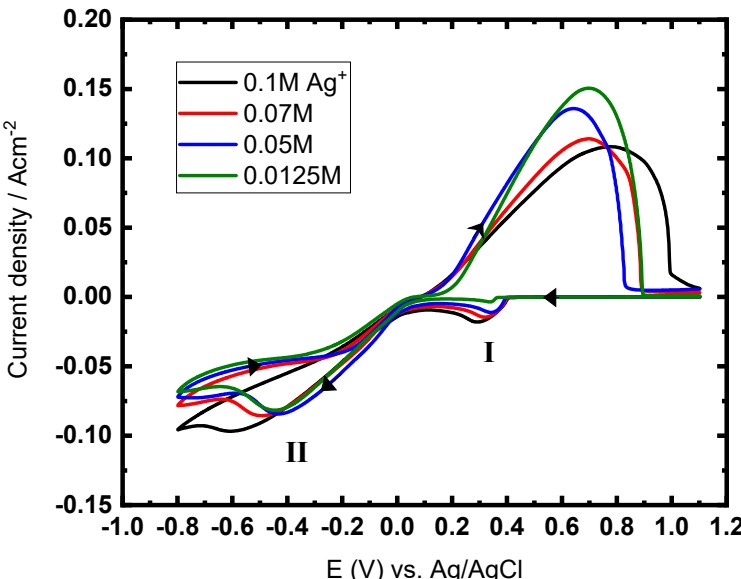

**Figure 4.** Typical CVs for CuAg reduction with constant $Cu^{2+}$ ions concentration (0.5 M) and varying concentrations of $Ag^+$ ions, at GCE, 50 mV s$^{-1}$, and at 27 °C.

In the anodic linear stripping voltammetry (ALSV) experiment, the metal was deposited potentiostatically (at a certain potential) at the GCE for a given time [44]. Then, the potential was shifted linearly to the higher anodic potential in the same solution (in situ) at a scan rate of 5 mV s$^{-1}$ after that time. For the purpose of comparison, ALSV experiments on parent metal deposits (Ag and Cu) at a constant deposition duration of 100 s are shown in Figure 5. When compared to Cu, Ag has a comparatively low stripping charge, and it has a slightly less positive potential. These findings support the hypothesis that a copper-rich alloy will be formed from these baths. The CuAg has only one oxidation peak, indicating that the copper and silver contents of the alloy oxidized simultaneously. However, the stripping charge (the area beneath the peak) can be used as a quantitative indicator of cathodic current efficiency. The current efficiencies of metal deposition are qualitatively proportional to these quantities of electrical charge. A series of ALSV curves for the alloy co-deposited at a constant concentration of $Cu^{2+}$ ions (0.5 M) and at different concentrations of $Ag^+$ ions were carried out at a constant deposition potential ($-0.2$ V) with a deposition time of 100 s, as shown in Figure 6. Even though their standard potentials are far apart, the two components of the alloy (copper and silver) dissolve simultaneously. It is obvious that as the $[Ag^+]/[Cu^{2+}]$ ratio decreases, the amount of electricity used at the potential of the anodic peak increases, implying that as this ratio is reduced, current efficiency improves.

### 3.3. Characterization of the ncCuAg

3.3.1. Surface Characterization

The morphological details of the CuAg on the SS surface are shown in Figure 7. The deposition was achieved for 10 min at various current densities at room temperature in order to examine the variation of the surface covering as current densities changed. All the deposits obtained exhibited a spherical shape morphology typical for copper, since the alloy is rich in copper. According to Figure 7a–d, as the current density increased, the coating

coverage increased as well. Coverage is patchy and the deposited CuAg can be observed between the SS surface underneath at a very low current density (1.14 m Acm$^{-2}$) (Figure 7a). The reason for this is that insufficient amounts of copper ions are being transported to the substrate to allow for appropriate spherical nucleation and development. The flux of copper ions towards the cathode is large at higher current densities ($\geq$1.72 mA cm$^{-2}$, which promotes greater nucleation and improved surface coverage and results in a compact and continuous covering (Figure 7b–d). However, increasing the current density leads to a greater grain size in agreement with the results obtained from the XRD data (as shown later). However, at the constant current density, increasing the time of deposition has no significant change in morphology (Figure 7e,f). Figure 7g shows the ED of pure copper on the SS for comparison. Finally, a cross section was shown in Figure 7h, which illustrates that the thickness of the alloy coating could be in the range of 5.397–8.094 μm.

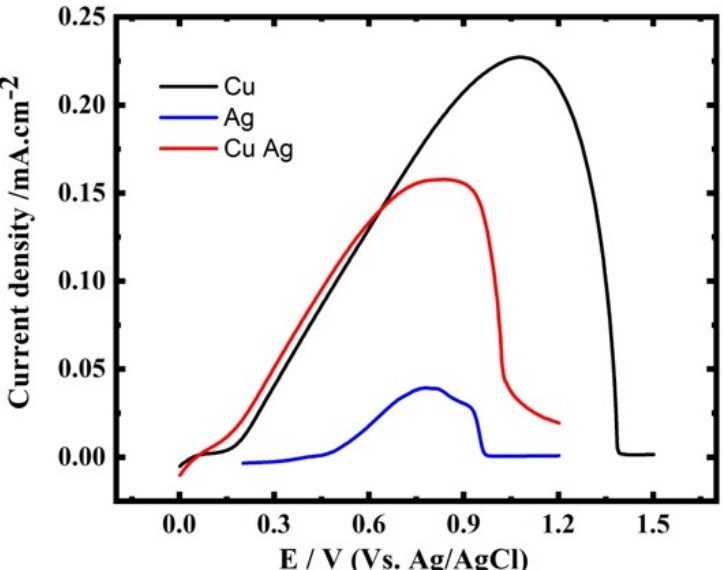

**Figure 5.** The ALSV curves for Ag, Cu, and CuAg at the GCE, scan rate 5 mV s$^{-1}$, and at 27 °C.

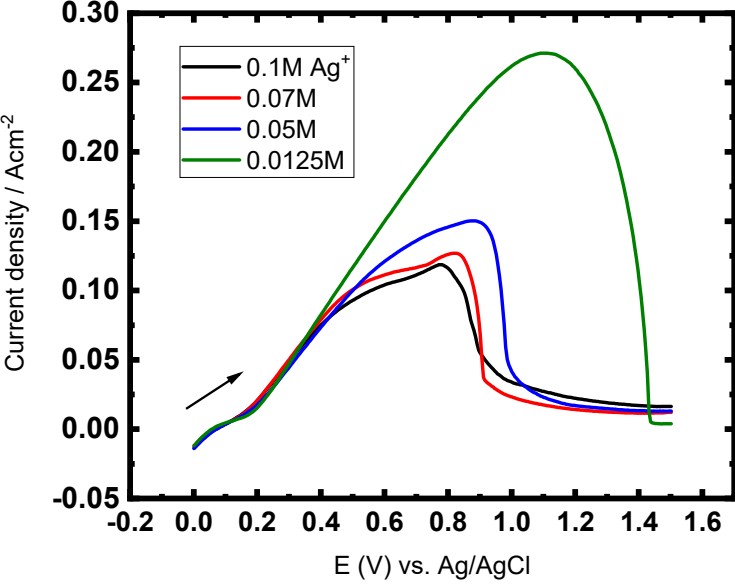

**Figure 6.** The ALSV recorded at GCE for CuAg with constant Cu$^{2+}$ ion concentrations and varying Ag$^+$ ion concentrations.

The mapping of the coated samples showing the presence and distribution of Cu, Ag, and an alloy of them is illustrated in Figure 8. However, Figure 9 shows some representative data of the ncCuAg analyzed using EDS. The ncCuAg co-deposited from the optimum bath I at different operating conditions of current densities, durations, and temperatures contains an Ag wt.% range of 2.17–6.15%. According to a review of the literature, adding a little amount of Ag to the Cu lattice improves copper's characteristics [37]. For example, the deposition of 4.0 percent silver with copper significantly increased the mechanical hardness and corrosion resistance when compared to a pure copper coating without causing significant electrical conductivity degradation [30].

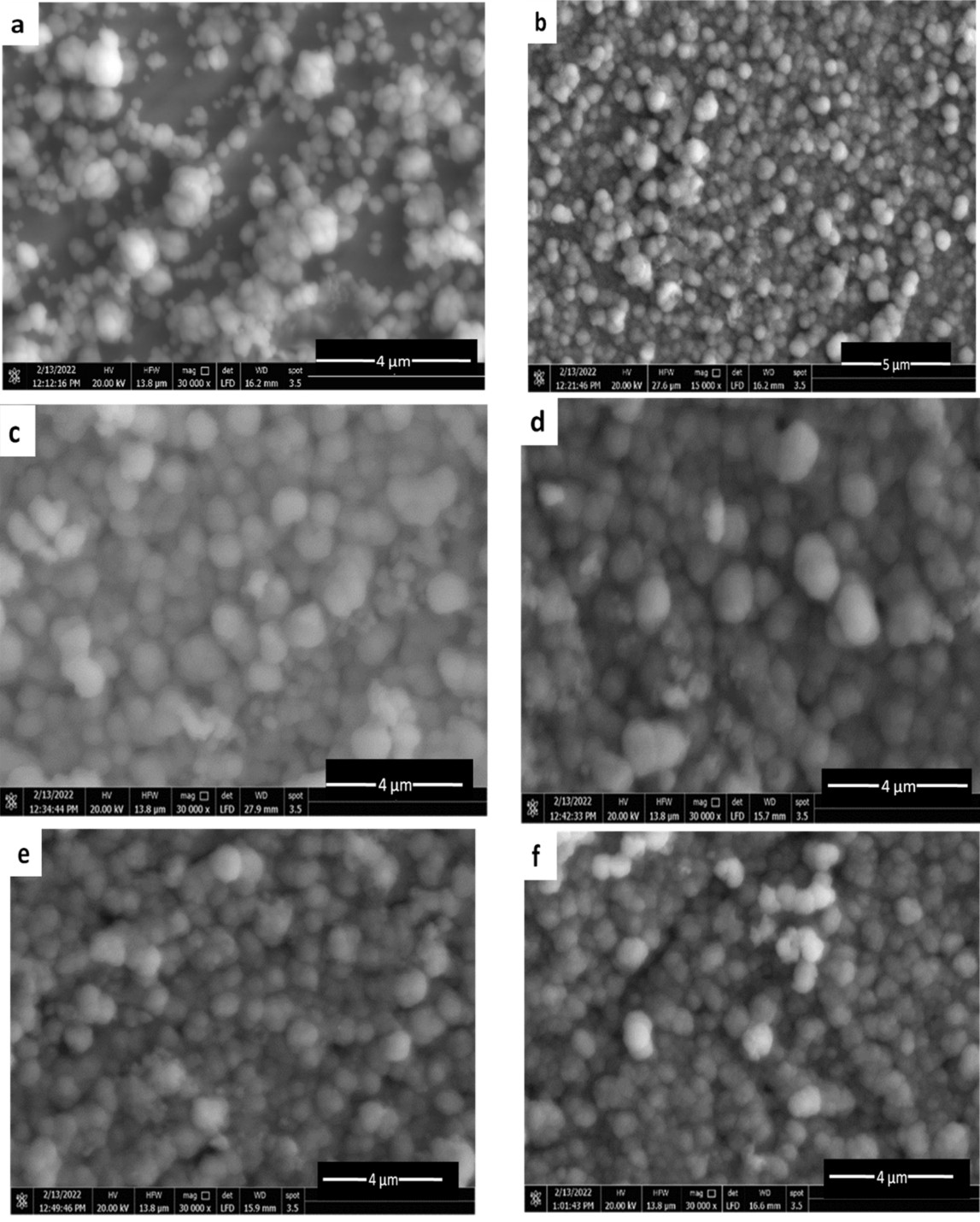

**Figure 7.** *Cont.*

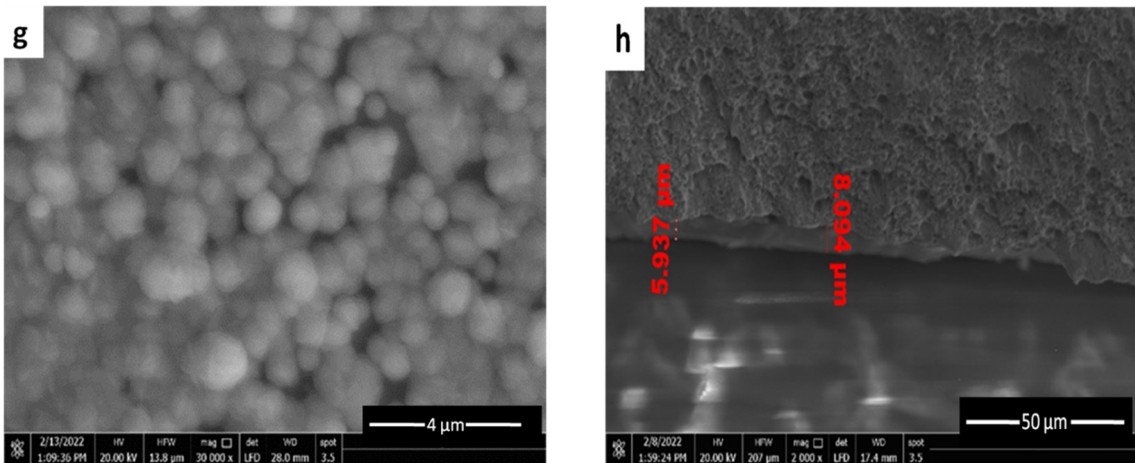

**Figure 7.** SEM of ncCuAg co-deposited from bath I at different operating conditions: (**a**) 1.14 mA cm$^{-2}$, 10 min; (**b**) 1.72 mA cm$^{-2}$, 10 min; (**c**) 2.29 mA cm$^{-2}$, 10 min; (**d**) 3.43 mA cm$^{-2}$, 10 min; (**e**) 1.72 mA cm$^{-2}$, 15 min; (**f**) 1.72 mA cm$^{-2}$, 20 min; (**g**) pure Cu, 1.72 mA cm$^{-2}$, 10 min; (**h**) cross section (1.72 mA cm$^{-2}$, 10 min).

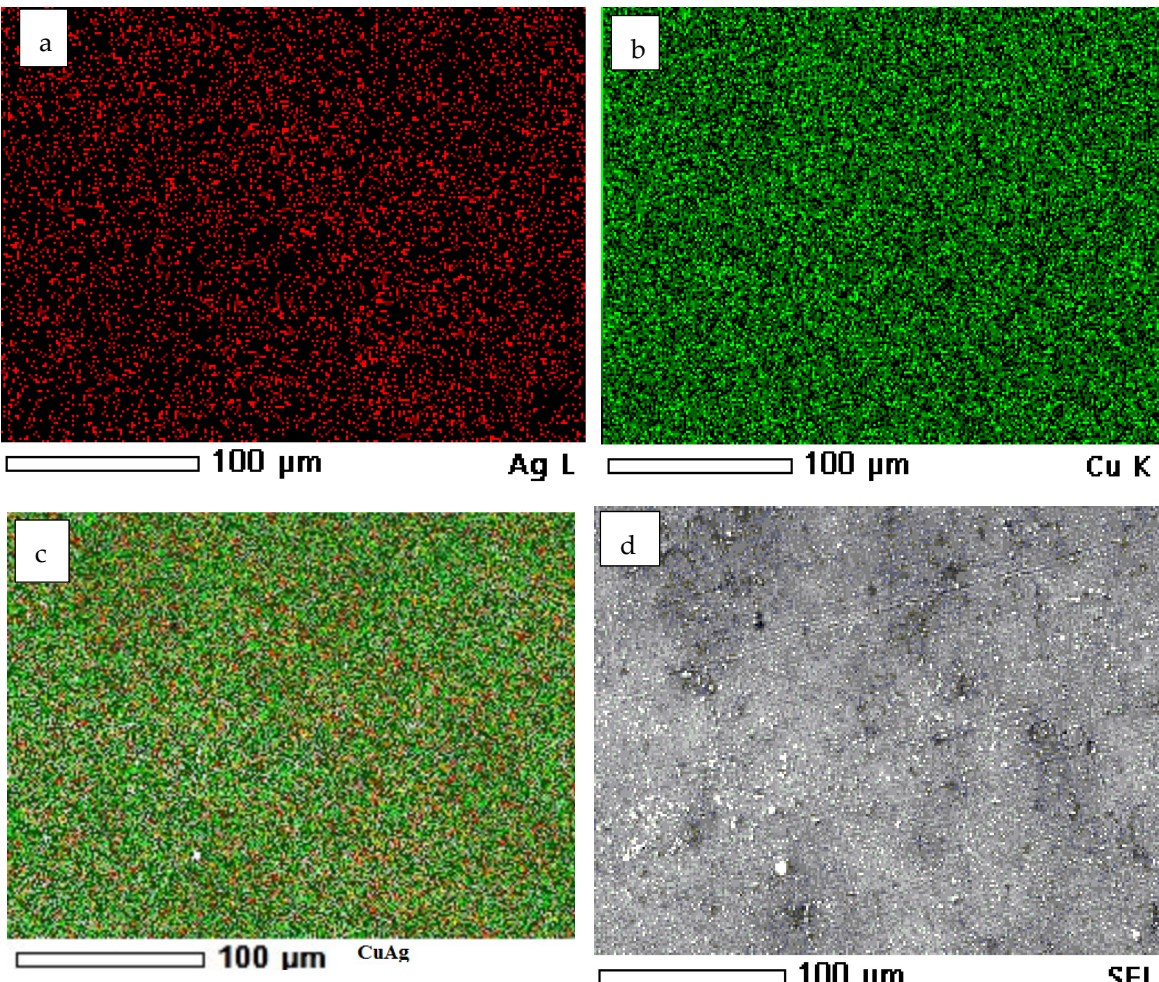

**Figure 8.** Mapping showing the distribution of the elements: (**a**) Ag; (**b**) Cu; (**c**) CuAg; and (**d**) original sample.

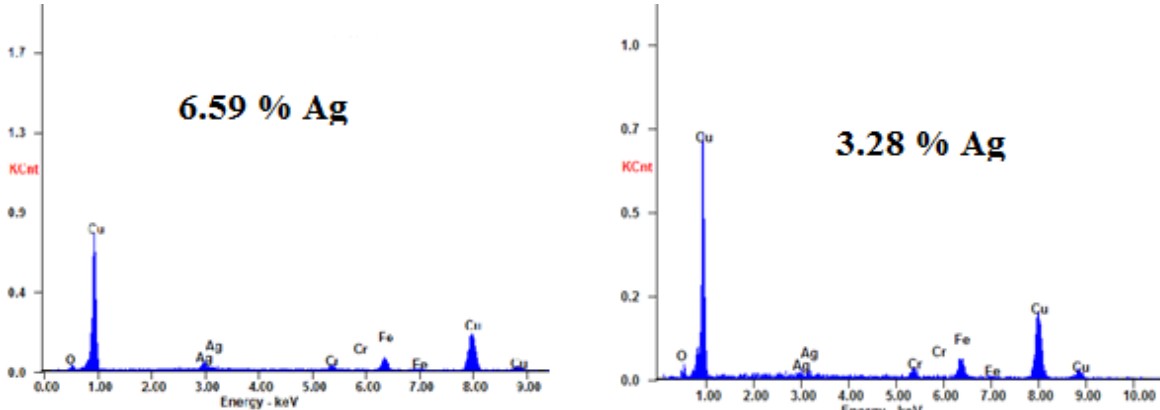

**Figure 9.** EDS of ncCuAg sample deposited from bath I at 1.72 mA cm$^{-2}$, (**left**) and at 2.29 mA cm$^{-2}$, (**right**) 10 min, and at 27 °C.

The HRTEM (high resolution transmission electron microscope) is the easiest and most reliable method for determining the size of metal nanoparticles, as it can reveal not only their size and shape, but also their crystalline structure. By using a TEM, the alloy nanoparticles generated by the ED technique were evaluated for homogeneity and particle size. The TEM examination results in Figure 10 show that the produced ncCuAg is composed of a spherical shape. Most of the ncCuAg particles were found to be spherical, with an average diameter of 4.10–6.12 nm.

SAED (selected area electron diffraction) (Figure 10d) is a qualitative analysis method of crystal structures from a spot diffraction pattern, which is obtained from the illumination of a parallel electron beam on a specimen. When entering a selector (chosen-region) slot into the image level of the objective lens, a deviation pattern is obtained from a sample area of a random 100 nm diameter. This method enables us to identify the lattice type, lattice parameters, and crystallographic orientation of this selected area. To analyze patterns of SAED, we integrate the geometric relationship and Bragg's equation in the reciprocal space. The observed Debye-Scherer rings are completely enclosed, indicating the CuAg nanostructure is highly crystalline in nature. The rings change from continuous to dotted as the size of the polycrystalline grains increases. By calculating the d-values (the spacing between lattice planes) and by comparing this value with the d-value of different phases of silver and copper in literature, we can identify the type of crystal lattice. Meanwhile, the crystalline samples will result in bright spots if the sample is polynanocrystalline (small spots making up rings, each spot arises from the Bragg reflection from an individual crystallite). Figure 10d shows the particle size distribution of the optimum samples of Cu-Ag. It was found that the Cu-Ag alloy size was in the range of 4.10–6.12 nm.

Figure 10e depicts its size distribution histograms. The histogram shows an average particle size of 6.0 ± 1 nm. Due to the creation of polycrystalline aggregates, the crystallite size of the particle differs from the particle size [45].

The main purpose of the XRD measurements was to study and clarify any possible nano-alloy formation between Cu and Ag, in which one single metallic phase should predominate and the other, if present, is indistinguishable. It is also possible to observe how the relative abundance of bimetallic compositions affects the product's crystallinity and chemical stability. XRD analysis was used to evaluate the CuAg coatings co-deposited at varying current densities (Figure 11). First, the appearance of fine and intense planes of (111), (200), (220), (311), and (222) indicates that the CuAg-coated samples have high crystallinity, which is consistent with the XRD data obtained for copper deposited from cyanide baths. Following JCPDS "00-150-9079", the individual samples are indexed to the cubic structure of space group Fm3m. These four preferred orientations (intense peaks) match a previously reported XRD diffraction result of copper with a purity of >99.999 percent (JCPDS, 04-0783) [46]. Except for the sample that was manufactured at a

higher current, the favored orientation plane of the other CuAg-coated samples is (311), which was deposited at 3.4 mAcm$^{-2}$. With increased current density, the intensity of the other peaks also increases. The diffraction peak is broadened by crystal imperfections. The diffraction peaks broaden as deviations from perfect crystallinity spread indefinitely in all directions. The amorphousness of the material is enhanced by the interstitial distribution of Ag within the Cu lattice [47]. The Scherrer Equation (3) is widely used to compute the crystal size of XRD data for each hkl plane. The crystallite size (D) can be determined using the following formula:

$$D = \frac{0.9\,\lambda}{\beta\,\text{COS}\,(\theta)} \tag{3}$$

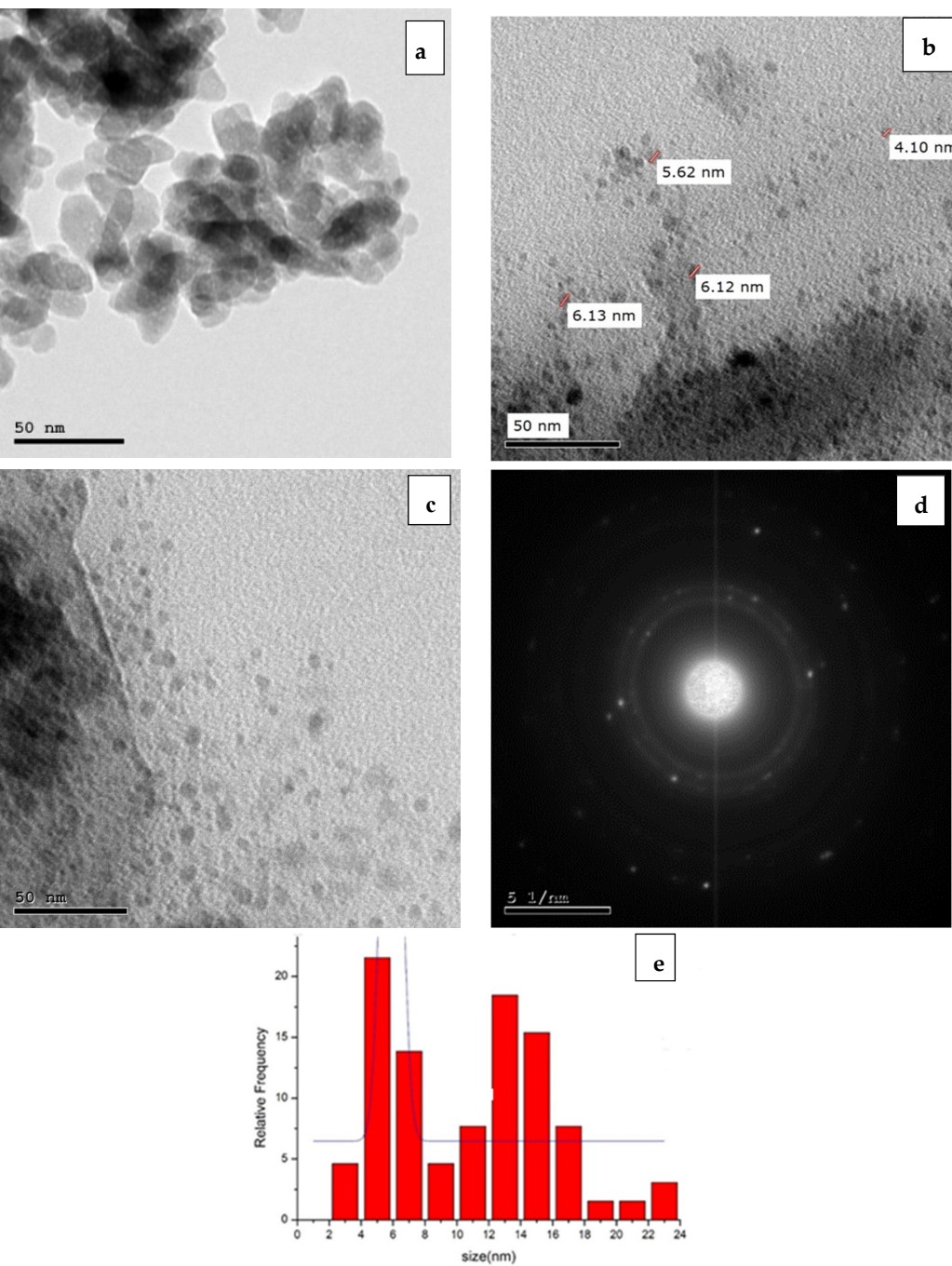

**Figure 10.** TEM images for ncCuAg electrodeposited on SS surface from the optimum bath at 1.72 mA cm$^{-2}$, 10 min, 27 °C, and the histogram of particle size distribution.

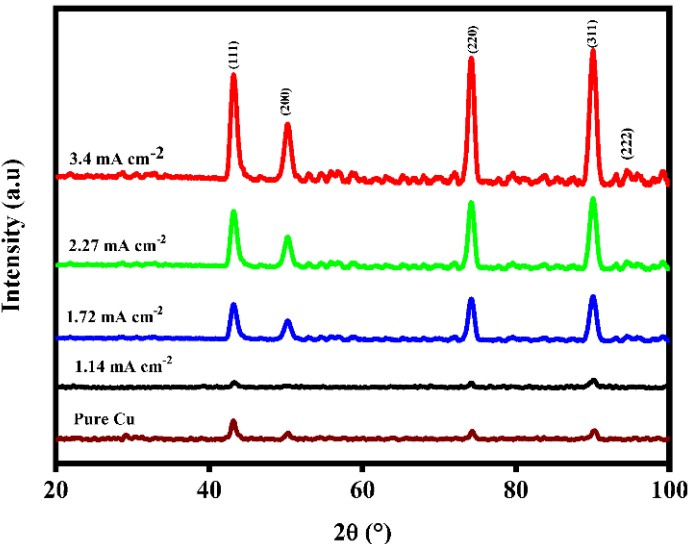

**Figure 11.** XRD patterns of ncCuAg.

Where D is the average crystallite size, λ is the wavelength of the x-ray and takes 1.54 Å for $CuK_\alpha$, β is the peak width of the strongest diffraction peak at half maximum height (FWHM) in radians (corrected for the instrumental broadening), and $\theta_{hkl}$ is the diffraction angle of the crystal plane (hkl). Table 2 shows the values of the average crystallite size [48]. Meanwhile, the microstrain (ε) (Table 3) calculated by Williamson Hall plots from the following equation [49–51]:

$$\beta_{hkl}\cos\,\theta = \frac{(k \times \lambda)}{D} + 4\varepsilon\,\sin\,\theta \tag{4}$$

**Table 2.** Crystallite size of the ncCuAg determined using XRD analysis.

| CuAg Alloy Coatings Codeposited at 27 °C | D (nm) ± 5 |
|:---:|:---:|
| 1.14 $mAcm^{-2}$, 10 min | 13.5 |
| 1.72 mA $cm^{-2}$, 10 min | 16.5 |
| 2.29 mA $cm^{-2}$, 10 min | 16.6 |
| 3.43 $mAcm^{-2}$, 10 min | 13.9 |
| 1.72 $mAcm^{-2}$, 15min | 13.4 |

**Table 3.** The microstrain values of the different orientation.

| Orientation | Microstrain (ε) |
|:---:|:---:|
| (111) | 0.00442 |
| (200) | 0.00651 |
| (220) | 0.00514 |
| (311) | 0.00322 |
| (222) | 0.00351 |
| Cu pure | 0.0037 |

By alloying with silver, the predicted microstrain of the copper matrix increases (Table 2). It is well known that XRD analysis gives the average crystallite size, while the TEM analysis gives the particle size.

XPS measurements have been carried out to ascertain the chemical composition and oxidation state of the ncCuAg. The surveyed X-ray photoelectron spectrum for the CuAg coating shows the presence of elemental Ag, Cu, O, and C in the sample (Figure 12a). The binding energies of the $Ag3d_{3/2}$ and $Ag3d_{5/2}$ orbits observed in the HR spectrum are 373.8 and 367.8 eV, respectively, indicating that the Ag is present as Ag° [52]. On the other hand, the binding energies of the $Cu2p_{1/2}$ and $Cu2p_{3/2}$ orbits are 951.84 and 931.02 eV, respectively, referring to the presence of Cu° [53]. Thus, the Ag and Cu atoms exist as metallic elements in the CuAg, indicating an improved resistance of the CuAg to oxidation [54].

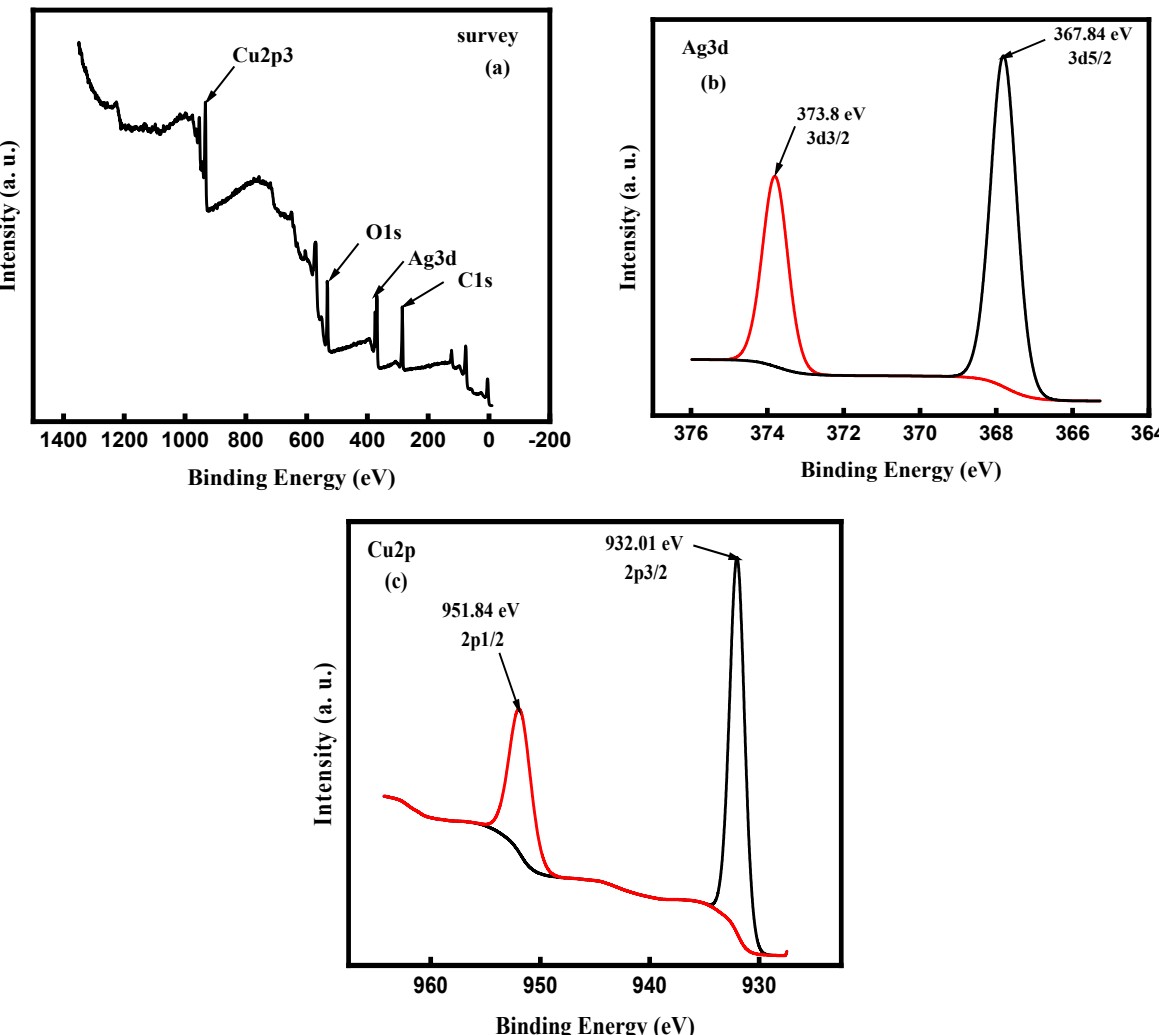

**Figure 12.** XPS results of the as-synthesized ncCuAg (**a**) Survey spectra, (**b**) Ag3d spectra, and (**c**) Cu 2p spectra.

### 3.3.2. Adhesion Test of the ncCuAg Coating

The cross-hatch-method was used as a method to measure the adhesion of the ncCuAg coating on the SS surface prepared using the optimum bath I, at 1.72 mA cm$^{-2}$, 10 min, pH 1.9, and at 27 °C. Figure 13 shows the CuAg coat before and after the measurement. A cheap cross-hatch cutter test kit makes this test quick and easy. Using a cross-hatch cutter, a lattice pattern is sliced into the finish film all the way to the substrate. The remaining loose film finish particles are removed by brushing the test area five times diagonally in each direction.

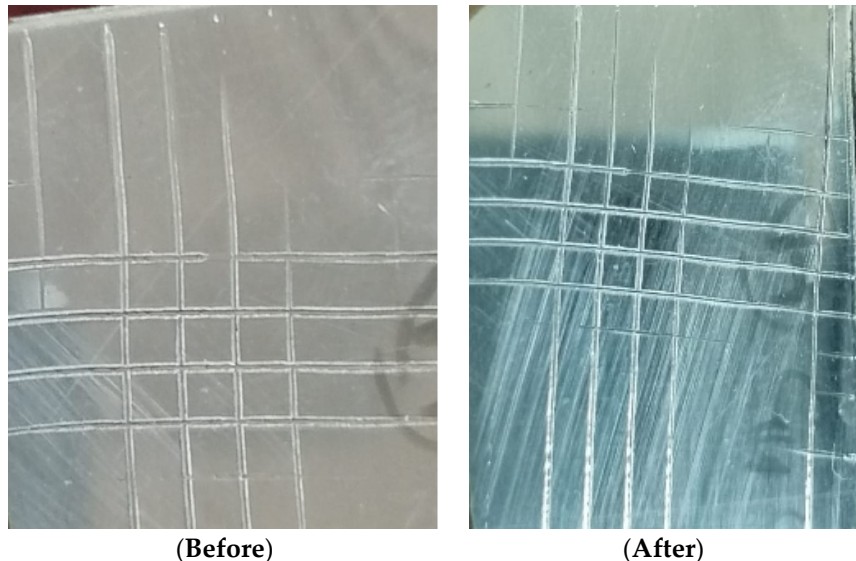

(**Before**)          (**After**)

**Figure 13.** The cross-hatch method for measuring the adhesion of ncCuAg.

As demonstrated, all of the cut edges are totally smooth, and no lattice squares are coming loose. The result of the test shows that the measured sample belongs to ISO class 0/ASTM class 5B. This means that the ncCuAg co-deposited under this operating condition from the studied bath exhibits excellent adhesion on the SS surface. Moreover, the pull-off technique was employed to estimate the quantitative adhesion force of the ncCuAg deposits. At the following working parameters: pH 1.9, at 1.72 mA cm$^{-2}$, 10 min, and at 27 °C, an adhesion force of $1.7 \times 10^7$ Pa was obtained.

### 3.3.3. Antibacterial Activity

Results presented in Table 4 revealed that by comparing with samples of copper only, copper silver samples at different conditions of deposition have a significant antibacterial effect on both tested gram-positive (*Staphylococcus aureus*) and gram-negative (*Escherichia coli*) bacteria, but the effect was stronger in the case of gram-negative than gram-positive bacteria. Among the prepared alloys, the copper silver prepared at 1.72 mA cm$^{-2}$ for 10 min was the most potent in inhibiting bacterial growth (Figure 14). In the present study, the fabrication cost is at an affordable level since an antibacterial effect of ncCuAg with a low Ag% (content up to 6.5%) could be achieved.

**Table 4.** The inhibition zone diameters of samples prepared at different operating conditions.

| Samples Prepared from Bath S under the Following Conditions | | Inhibition Zone Diameter (mm) | |
|---|---|---|---|
| | | **Gram (+ve) Bacteria** *Staphylococcus aureus* **(ATCC 6538)** | **Gram (−ve) Bacteria** *Escherichia Coli* **(ATCC 8739)** |
| Blank (uncoated SS) | | NA | NA |
| **Copper only** | | **12** | **NA** |
| **CuAg alloy** | (1.14 mAcm$^{-2}$, 10 min) | 30 | 30 |
| | (1.72 mA cm$^{-2}$, 10 min) | 38 | 46 |
| | (2.29 mA cm$^{-2}$, 10 min) | 29 | 33 |
| | (3.43 mAcm$^{-2}$, 10 min) | 40 | 42 |
| | (1.72 mAcm$^{-2}$, 15 min) | 35 | 41 |
| | (1.72 mA cm$^{-2}$, 20 min) | 32 | 36 |
| | (1.72 mA cm$^{-2}$, 10 min, 35 °C) | 27 | 40 |
| | (1.72 mAcm$^{-2}$, 10 min, 45 °C) | 29 | 38 |

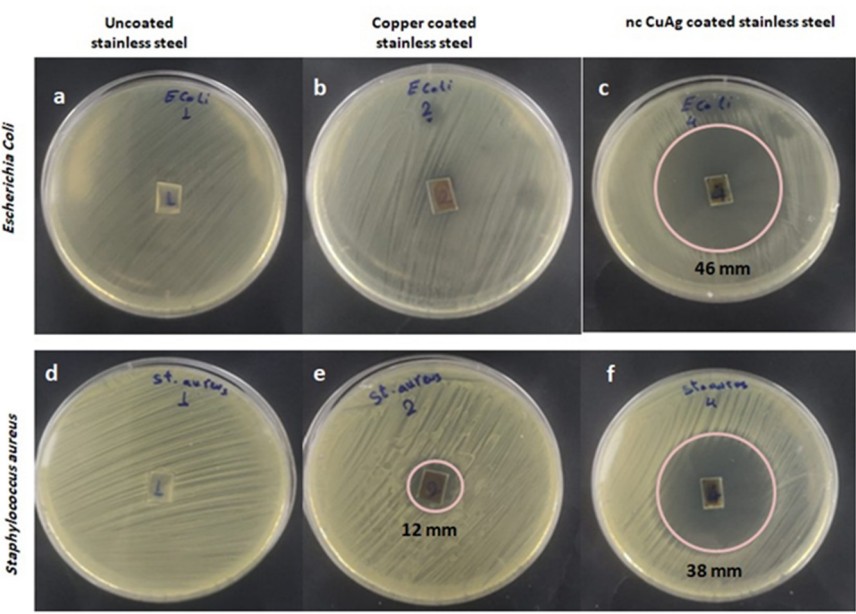

**Figure 14.** Antibacterial effect of uncoated stainless steel (**a**,**d**), copper-coated stainless steel (**b**,**e**), and the ncCuAg-coated stainless steel (**c**,**f**) against *Escherichia coli* (**a**–**c**) and *Staphylococcus aureus* (**d**–**f**).

Copper is thought to cause microbial death, but the exact mechanism and relative importance of each mechanism are unclear. One method involves the physical interaction of the CuNPs with the cell membrane or viral plasma membrane, leading to its destruction, thereby making the microbe more susceptible to damage from copper ions [55]. A second mechanism for copper's action is its ability to generate reactive oxygen species (ROS) through a Fenton-like reaction, leading to lipid peroxidation, protein oxidation, and DNA damage caused by enzymes and non-enzyme mediated oxidative damage [56,57]. A copper surface has been demonstrated to entirely eliminate *MRSA* and *Escherichia coli* in just a few hours [58]. Ag and Cu components are mainly used as alloying elements. Both of these elements have been reported to have broad spectrum antibacterial activity. Due to the release of metal ions, it has been demonstrated that surface coatings containing Ag or Cu can resist the adhesion of cells, colonization, and biofilm formation [59,60]. Recent research on antibacterial metal alloys has also demonstrated that Cu- and Ag-containing alloys have anti-adhesion and anti-biofilm functions due to the metal ion release [61,62]. The current work demonstrated that the addition of silver to copper in an alloy raised its antibacterial effect more than copper alone, which indicates the higher efficiency of silver as an antimicrobial agent. A silver ion or silver-based compound can cause significant toxic effects on microorganisms [63], demonstrating biocidal effects on up to 16 species of bacteria [64]. Ag nanoparticles can disrupt and impair different cellular and metabolic pathways via nonoxidative and oxidative mechanisms [65]. The positive charge of silver nanoparticles strongly interacted with the membrane's negative charge of bacterial cells when they became in contact with each other, making it easier for the membrane to cling to the cell wall. After adhering to the cell wall, some of these particles disintegrate into physiologically active $Ag^+$, which links to the cell wall, releasing additional $Ag^+$ ions [66]. These free ions result in changes in the membrane structure and cell wall disruption by interaction with sulfur-containing proteins, forming many gaps in the cell wall, which leads to alteration of the membrane permeability and vital cellular content losses [66]. Moreover, silver nanoparticles also cause the denaturation of important cellular molecules such as DNA and proteins. As a result of the aforementioned effects of both Cu and Ag nanoparticles, the alloys prepared from both elements were more efficient than those of Cu alone. As for the emergence of a stronger antibacterial effect in the case of gram-negative bacteria than in gram-positive bacteria, this is probably because gram-negative bacteria had a thinner cell wall and were more sensitive to damage. Antibacterial stainless steel is

now being developed with a focus on *Escherichia coli* and *Staphylococcus aureus*, the most common causes of implant-associated infections, as reported by Nan et al. [67].

## 4. Conclusions

A highly adherent nanocrystalline copper-silver (ncCuAg) on a stainless-steel substrate was successfully synthesized using a novel plating bath, free from cyanide. It was found that an ionic ratio $[Ag^+]/[Cu^{2+}]$ in the range of 0.025–0.2 produces a high-quality ncCuAg with Ag content ranging from 2.17 to 6.59% (wt.%), depending on the operating conditions. The coating samples at different conditions of deposition had a significant antibacterial effect on *Staphylococcus aureus* and *Escherichia coli*, but the effect was stronger in the case of *Escherichia coli* than in *Staphylococcus aureus*. The crystallite size calculated from the X-ray peak broadening was found to be in the range of 13.4–16.6 nm, and the increasing current density leads to a greater grain size. By alloying with silver, the predicted microstrain of the copper matrix increases. Both HRTEM, FESEM-EDS, XPS, and mapping analysis revealed that the ncCuAg is composed of both Ag and Cu atoms. Electrochemical processes occurring during the CuAg co-deposition were investigated by using the linear sweep voltammetry (LSV), cyclic voltammetry (CV), and anodic linear stripping voltammetry (ALSV).

**Author Contributions:** Conceptualization, M.A.E.S. and M.A.M.I.; methodology, N.T.E.; software, M.G.; validation, M.A.E.S., M.A.M.I. and N.T.E.; formal analysis, M.G.; investigation, N.T.E.; resources, M.A.M.I.; data curation, N.T.E.; writing—original draft preparation, M.A.M.I.; writing—review and editing, M.A.M.I.; visualization, M.G.; supervision, M.A.M.I.; project administration, M.A.E.S. All authors have read and agreed to the published version of the manuscript.

**Funding:** This research was funded by Qassim University, grant number 10089-cosab-bs-2020-1-3-I.

**Data Availability Statement:** Not applicable.

**Acknowledgments:** The authors gratefully acknowledge Qassim University, represented by the Deanship of Scientific Research, for the financial support for this research under the number (10089-cosab-bs-2020-1-3-I) during the academic year 1442 AH/2020 AD.

**Conflicts of Interest:** The authors declare no conflict of interest.

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
