# Peer review of "Electrochemical Synthesis of Nanocrystalline CuAg Coatings on Stainless Steel from Cyanide-Free Electrolyte"

_processes, doi:10.3390/pr10102134_

Round 1

Reviewer 1 Report

Dear Authors, 

The article submitted for review presents the physical and chemical properties of the Cu-Ag coating formed on stainless steel by the electroplating method. This type of solution can find application wherever there is a risk of contact with various types of bacteria. 

The manuscript in its current form needs improvement on many levels. The comments that come to my mind after reading the text several times I can divide into several groups. Language issues: the level of English is acceptable in many places. However, you could try (especially at the beginning of individual paragraphs) to edit subordinate complex sentences. Shorter sentences will improve the readability of the entire manuscript. As it stands, there are a lot of paragraphs that can be defined as hard-to-read. The title of the article should also undergo such treatment. In my opinion, it is too long (the word "alloy", and the mention of biological activity etc. should be removed).  Certain sublime phrases can be replaced with simpler forms of expression: 'by means of'>'using'; 'for the purpose of'>'for'; 'in accordance with'>'by/under'; 'in order to'>'to'; 'is thought to be'>'is considered'; 'in the vicinity of'>'near'; 'at the completion of'>'after'; 'have been shown to have the ability to'>'can'.... In this way, the creation of so-called wordy sentences is avoided. In addition, You need to check the correctness of the use of specific words in the manuscript (Not all words are used in the right context). In a few cases, I can suggest changing specific phrases, for example: 'stainless-steel'>'stainless steel'; 'codeposition'<'co-deposition'; 'effectiveness'>'effects'; 'also'>'moreover'; 'more noble'>'nobler'; 'diffusion limited'>'diffusion-limited; 'varies'>'varying'; 'resulting'>'results'; 'quantity'>'amount'; 'arising'>'arises'; 'd-vale'>'the d-value'. The text contains a large number of words which stuck together. Since the numbering of the verses is not marked, I can not show precisely where I noticed this kind of problem. In addition, it makes it difficult for me to point out places where commas are missing, incorrect prepositions are used, and articles that occur can be changed. I believe that certain abbreviations should not be used, i.e. words should be written in full form: ''It's'>'It is''; 'E.coli'>'Escherichia coli'. I think it is important to look meticulously at whether there is a correlation between the grammatical tense used and the form of the noun. 

The methodological issues: The use of the word 'alloy' should be sorted out. It is naturally ascribed to the situation when the material was obtained by processes of fusion of one metal with another. In the situation described in the manuscript, You should stop using the phrase 'CuAg coatings'. 

Why was a NaOH solution used in the process of obtaining the layer? Was the use of another precipitating agent, such as ammonia solution or ammonium carbonate solution, considered? 

How can you be sure that the pH during this process had a constant value and was specifically 1.9? From my point of view, a realistic pH range should be given. In the manuscript ther is no explaination the reasons for using silver nitrate(V) solution in such a wide range. 

Did the authors consider the co-deposition process from the point of view of a simple sol-gel synthesis? The ingredients used during the synthesis offer the possibility of obtaining in parallel still other products like hydroxides of the corresponding metals. Is it known with certainty what other products led to the Cu-Ag coating?

There is an error in the notation of reaction 2 by not agreeing on the stoichiometry. In the selected figures (Fig. 1; 3; 5), the word 'alloy' should be removed as this is a kind of overinterpretation. The scale on the TEM microphotographs is blurred, and the size of the Cu-Ag particles cannot be read. Analysis of the adhesion and cohesion of the layer should be carried out using the so-called scratch test. It should be determined at what force the layer detaches and at what force it cracks. The adhesion measurements presented in the manuscript, together with the drawing, do not contribute any relevant information. And the abstract states that tests were performed on various gram-positive and gram-negative bacteria, while the results are only for Staphylococcus aureus and Escherichia coli. I think that both the abstract and the summary should be re-edited in terms of really showing what was done and the value of this work. The description of the effect of silver or copper particles on bacteria, etc. posted after Fig. 14 has no connection to the results. It is like a part of the theoretical introduction and does not explain what the authors got in their work. In addition, the posting of the determination of the most likely number of bacteria is also missing here. This measurement is very simple and is a natural extension of the topic of testing the antibacterial properties of the tested surface. From my perspective, this is a significant deficiency that calls into question the validity of providing any antibacterial measurements.

Best regards,  

Reviewer

Author Response

Dear reviewer #1

Reviewer 2 Report

The focus of this study is to prepare a highly adherent nanocrystalline copper-silver (ncCuAg) alloy coating on a stain-less-steel substrate using a novel plating bath, free from cyanide, and to investigate their antibacterial effects. Overall, the article is logical and relatively informative. However, the writing skill shown in this article still needs to be improved, some details need to be more clear explanation and supplements and the structure of the article needs to be optimized. Therefore, it also needs to be modified from the following points:

 1.Please delete the border of the figures in the article. For example, Figure 7 and Figure 8.

2.The font and format of the formula in this article should be consistent, for example, Formula 3 and Formula 4.

3.In section 3.3.4, Please provide relevant reaction equations, figures and data, because the author's description alone lacks the concise and vivid explanation, which could confuse the readers.

4.Please optimize the structure and size of figures in this article. Many charts are inconsistent in size, not aligned and have lots of white space. For example, Figure 10 and Figure 12.

5.Section 3.2.1, section 3.2.2 and section 3.2.3 can be merged together.

Author Response

Dear reviewer 2

Round 2

Reviewer 1 Report

Dear Authors, 

After modifications the manuscript has better quality than its previous version. I accept the explanations provided. I am pleased that the essential part of my comments has been implemented. From the perspective of a person who has read the text carefully several times, I would like to say that the manuscript is more understandable. I still found a few minor issues that could be improved. The therm 'to combat germs' could be replaced with 'bacteria control' or something similar. The proposed expressions is used more in another field, namely, in computer games. In addition, beware of using words that are overly emotional and thus can have a difficult to define meaning, e.g. 'excellent'. There is no reference to formula (4) in the manuscript. Not all microstrain values have the same accuracy. 

Sincerely yours,

 Reviewer

Author Response

Dear Prof. Dr. reviewer 1
